# Differences in Placental Imprinted Gene Expression across Preeclamptic and Non-Preeclamptic Pregnancies

**DOI:** 10.3390/genes11101146

**Published:** 2020-09-29

**Authors:** Maya A. Deyssenroth, Qian Li, Carlos Escudero, Leslie Myatt, Jia Chen, James M. Roberts

**Affiliations:** 1Department of Environmental Health Sciences, Columbia University, New York, NY 10032, USA; 2Department of Environmental Medicine and Public Health, Icahn School of Medicine at Mount Sinai, New York, NY 10029, USA; qian.li@mssm.edu (Q.L.); jia.chen@mssm.edu (J.C.); 3Group of Research and Innovation in Vascular Health (GRIVAS Health), Chillán 4081112, Chile; cescudero@ubiobio.cl; 4Department of Basic Sciences, Faculty of Sciences, Universidad del Bío-Bío, Chillán 4081112, Chile; 5Department of Obstetrics and Gynecology, Oregon Health and Science University, Portland, OR 97239, USA; myattl@ohsu.edu; 6Magee-Womens Research Institute, Department of Ob/Gyn and Reproductive Sciences, Epidemiology and Clinical and Translational Research, University of Pittsburgh, Pittsburgh, PA 15213, USA; jroberts@mwri.magee.edu

**Keywords:** preeclampsia, placenta, imprinted genes

## Abstract

Preeclampsia is a multi-systemic syndrome that presents in approximately 5% of pregnancies worldwide and is associated with a range of subsequent postpartum and postnatal outcomes, including fetal growth restriction. As the placenta plays a critical role in the development of preeclampsia, surveying genomic features of the placenta, including expression of imprinted genes, may reveal molecular markers that can further refine subtypes to aid targeted disease management. In this study, we conducted a comprehensive survey of placental imprinted gene expression across early and late onset preeclampsia cases and preterm and term normotensive controls. Placentas were collected at delivery from women recruited at the Magee-Womens Hospital prenatal clinics, and expression levels were profiled across 109 imprinted genes. We observed downregulation of placental Mesoderm-specific transcript (*MEST*) and Necdin (*NDN*) gene expression levels (false discovery rate (FDR) < 0.05) among early onset preeclampsia cases compared to preterm controls. No differences in placental imprinted gene expression were observed between late onset preeclampsia cases and term controls. While few studies have linked *NDN* to pregnancy complications, reductions in *MEST* expression levels, as observed in our study, are consistently reported in the literature in relation to various pregnancy complications, including fetal growth restriction, suggesting a potential role for placental *MEST* expression as a biosensor of an adverse in utero environment.

## 1. Introduction

Preeclampsia is a multi-systemic syndrome that affects women during pregnancy, with a worldwide prevalence of approximately 5% [1,2]. The syndrome is characterized by new onset hypertension and proteinuria or other systemic findings after 20 weeks’ gestation [3]. Signs and symptoms can vary greatly between individuals. Once these clinical features manifest, they are not reversible until the delivery of the placenta [4]. Although acute symptoms abate with delivery, women who experience preeclampsia continue to be at risk for adverse health outcomes extending beyond the pregnancy period, including cardiovascular disease [5] and possible cognitive decline [6,7]. Similarly, infants who are exposed to preeclampsia in utero are also at elevated risk for postnatal health outcomes over the course of life, including a predisposition toward elevated blood pressure [8], stroke [9], neurodevelopmental outcomes [10,11], and growth-restriction related metabolic syndrome [12].

Concerted efforts over the years to identify preclinical markers of preeclampsia have been frustrated by the inability to identify markers distinguishing women who go on to develop the disease from women who do not. This ambiguity in the literature suggests that the current classification of preeclampsia captures a heterogeneous set of disorders. For example, the clinical trajectory among cases with early onset preeclampsia (diagnosed at <34 weeks’ gestation) compared to cases with late onset preeclampsia (diagnosed at >34 weeks’ gestation) is known to differ [1,3]. Differences include uterine Doppler abnormalities that precede early onset but not late onset preeclampsia and a greater likelihood for fetal growth restriction among infants born to early onset cases compared to late onset cases [13]. The long-range impact is also more striking with early onset preeclampsia, with an almost 10-fold increased risk of cardiovascular disease-related death compared to less than double the risk with late onset preeclampsia [14]. Infant sex differences have also been observed, notably that early onset preeclampsia is more prevalent among pregnancies of female fetuses, suggesting that male fetuses may be more susceptible to pregnancy loss as a result of the compromised gestational state [15].

Abnormal placentation is considered the root cause of preeclampsia since signs and symptoms are present even when the fetus is absent, as in the obstetrical condition hydatidiform mole in which only placental tissue develops. Furthermore, clinical findings subside once the placenta is delivered [16,17]. While placental dysfunction is common to both early and late onset preeclampsia, the route to this trophoblastic stress differs. In early onset preeclampsia, abnormal remodeling of maternal vessels early in gestation results in placental malperfusion, subsequent hypoxia and inflammation, and ultimately triggers syncytiotrophoblast stress that leads to the onset of the characteristic maternal syndrome and fetal growth restriction [18]. In late onset preeclampsia, placental senescence in a previously normal placenta is proposed to lead to similar trophoblastic stress with the characteristic maternal findings, but less commonly with fetal growth restriction [18]. In both disorders maternal factors interact with the placental dysfunction to modify outcomes.

Various notable placental features are linked to preeclampsia presentation, including gross histological features [19] and circulating factors of placental origin [20,21]. Placental genomic patterns are also of interest, particularly imprinted genes, a subset of genes that are mono-allelically expressed on the basis of parent of origin [22,23]. These genes are highly expressed in the placenta and play a dominant role in regulating key placental processes [24]. Given their regulatory influence on placental development, imprinted genes have the potential to further characterize the underlying mechanism defining preeclampsia onset and differences in the clinical presentation of preeclampsia [25]. Indeed, several studies have explored placental imprinted features in relation to preeclampsia presentation in human [26,27] and animal studies [28]. In the current study, we build on this prior literature through a comprehensive survey of known imprinted genes to evaluate differences in expression, comparing placentas of women with early onset and late onset preeclampsia with those from normotensive women. We hypothesize differential imprinted gene expression among preeclampsia cases compared to controls, with a broader dysregulation observed among early onset cases.

## 2. Materials and Methods

Participants: Women were recruited at the Magee-Womens Hospital prenatal clinics from 2007 to 2012 to participate in the Prenatal Exposures and Preeclampsia Prevention: Mechanisms of Preeclampsia and the Impact of Obesity (PEPP3) study [29,30].

Women with singleton pregnancies and no known features predisposing them to preeclampsia other than obesity (i.e., without hypertension, diabetes, collagen vascular disease, or multiple gestations) were eligible for enrollment. The diagnosis of preeclampsia was based upon the criteria at the time of the study [31] and defined as new onset gestational hypertension (≥140 mm of mercury (mm Hg) systolic and/or 90 mm Hg diastolic after 20 weeks’ gestation accompanied by proteinuria (1+ on catheterized urine, 2+ on non-catheterized urine, 0.3 protein/creatinine ratio or 24 h urine of ≥300 mg/24 h)). Early onset cases were delineated from late onset cases using a 37-weeks-of-gestation cut point. This definition is motivated by the pathophysiological differences in the presentation of the disease, as preeclampsia is not associated with an increased risk of fetal growth restriction and there are fewer vascular changes in the placenta after 37 weeks of gestation. In the present analysis, the gestational age at delivery among early onset cases ranged from 30–35 weeks, with all but 2 deliveries occurring within 34 weeks’ gestation (and disease onset likely occurring prior to hospital admission). The gestational age at delivery among late onset cases ranged from 37–41 weeks. Hence, the early and late onset cases in the current study fall within the cut point definitions outlined by the American College of Obstetricians and Gynecologists (ACOG) [3]. Normotensive control subjects were women with similar exclusion criteria who did not develop preeclampsia, gestational hypertension, or growth restriction. Late onset preeclampsia cases were matched with normotensive controls based on maternal age, race, prepregnancy BMI, and gestational age at delivery. Due to the limited number of samples in our biobank from normotensive women delivering preterm, we were only able to match preterm controls with women with early onset preeclampsia based on maternal age and prepregnancy BMI. Given the severity and early presentation of disease, all early onset cases delivered prior to 37 weeks’ gestation while late onset cases delivered from 37 weeks onward. Accordingly, selected normotensive control subjects matched to early onset cases all delivered preterm and normotensive control subjects matched to late onset cases all delivered at term. In addition, given the parent study’s interest in obesity, the overall distribution of recruited participants was weighted toward overweight and obese women. All subjects gave their informed consent for inclusion before they participated in the study. The study was conducted in accordance with the Declaration of Helsinki, and the protocol was approved by the University of Pittsburgh Institutional Review Board (MOD08050339-24/PRO08050339).

Placental collection: Placentas were collected shortly after delivery, and four 2.5 cm full-thickness sections were obtained and were immediately snap-frozen in liquid nitrogen. Samples were then stored at −80 °C.

RNA extraction: RNA was extracted from placenta samples using the Maxwell simplyRNA Tissue Kit (#AS1280, Promega, Madison, WI, USA), following the manufacturer’s protocol. Extracted RNA was quantified using a Nanodrop 2000 spectrophotometer (Thermo Fisher Scientific Inc., Waltham, MA, USA) and stored at −80 °C until use.

Imprinted gene expression analysis: Placental RNA was profiled using a custom-designed nCounter code set containing 109 established and putative imprinted genes (nanoString Technologies, Seattle, WA, USA) as previously described [32]. Briefly, RNA was hybridized to reporter and capture probes, and purified target-dual probe complexes were aligned and immobilized on imaging cartridges using an nCounter Prep Station II. Cartridges were scanned in an nCounter Digital Analyzer for code count detection.

### Statistical Analysis

Data pre-processing: The raw nCounter code counts were normalized against the geometric mean of spike-in synthetic control probes included on each cartridge to account for variability introduced during cartridge preparation. To account for variability in sample input, code counts were additionally normalized against the geometric mean of the included housekeeping genes (*RPL19* and *RPLP0*). Sample-specific limit of detection (LOD) values were set at two standard deviations above the mean of the included negative control probes. For each sample, expression values that fell below the LOD were replaced with the LOD/sqrt2. Out of the 109 assayed imprinted genes, 92 genes were retained for further downstream analysis based on expression levels above the LOD in >50% of samples in at least one of the considered contrast groups (early onset cases, preterm controls, late onset cases, and term controls). One early onset case sample was considered an outlier on the basis of a visual Principal Component Analysis (PCA) inspection and removed from the dataset, leaving a final sample size of 99.

Differential gene expression analysis: Differential gene expression analysis was performed using the limma R package, and all models were adjusted for gestational age and race/ethnicity. Significant differences in gene expression were determined based on *p*-values adjusted for multiple comparisons using a false discovery rate (FDR) value of <0.05. These variables were selected based on observed distribution differences between the early onset cases and preterm controls. In addition to gestational age and race/ethnicity, birthweight also varied between early onset cases and preterm controls. However, due to its strong correlation with gestational age (Appendix A), birthweight was not included as a covariate to reduce model instability due to collinear predictors. The code implemented for this analysis is publicly available on GitHub (https://github.com/mdeyssen/CoLab). All analyses were conducted using R version 3.6.2 [33].

## 3. Results

There were no significant differences in demographic characteristics comparing late onset cases and term controls. Significant differences in gestational age and, concordantly, birth weight were apparent comparing early onset cases and preterm controls. While the overall study population was predominantly (>75%) white, the majority of preterm controls (68.4%) were non-white (Table 1).

Since the control groups selected to contrast against the early and late onset cases differed by gestational age, we evaluated the potential influence on imprinted gene expression due to this difference between the preterm and term controls; we observed no statistical differences (Appendix A). The sex of the fetus may also influence the molecular profiles of the placenta. Based on our panel of imprinted genes, we observed no differences in gene expression levels comparing male and female placentas among term controls (Appendix A).

Differential gene expression analysis comparing late onset cases and term controls revealed no differentially expressed genes (Figure 1A). Similarly, no differences were observed in the sex-stratified analysis (Figure 1B,C).

Comparing the early onset cases and preterm controls, we observed differential expression of the *Mesoderm-specific Transcript* (*MEST*) gene (Figure 2A) as well as the *Necdin* (*NDN*) gene (Figure 3A). Both *NDN* and *MEST* gene expression levels were downregulated in the early onset cases compared to preterm controls (Figure 3A and Figure 4A).

The significant downregulation of *MEST* found in the overall analysis was also observed in the sex-stratified analysis comparing early onset cases and preterm controls among female placentas (Figure 2B). Restricting the analysis to male placentas, no significant differences in expression were observed (Figure 2C). No sex-specific differences in *NDN* expression patterns were observed (Figure 2B,C and Figure 4B,C).

Finally, since the experience of labor during delivery may alter the molecular profile of the placenta compared to delivery without labor, we performed a sensitivity analysis removing four subjects who, unlike all other subjects included in this study, did not undergo labor. Significant associations and effect size estimates comparing cases and controls in the overall analysis were consistent with the observations in the labor-restricted analysis (Appendix A).

## 4. Discussion

Following a comprehensive profile of imprinted gene expression across early and late onset preeclampsia cases and matched normotensive controls, we observed downregulated *MEST* and *NDN* expression levels in the early onset cases compared to preterm controls. In vivo and in vitro studies evaluating *MEST* suggest that it plays a pivotal role in placental development, including angiogenesis [34] and invasion [35], processes known to be perturbed in preeclampsia. *NDN* is not known to play a direct role in placental dysregulation; however, its ubiquitous expression in neurons and role in neuronal differentiation highlight its importance early in development [36]. In adulthood, dysregulation of both *MEST* and *NDN* are implicated with the onset of obesity through adipose tissue expansion [37,38,39], suggesting potential postnatal health effects in energy balance regulation. *NDN* is additionally implicated in neurobehavioral outcomes [40,41]. Both *MEST* and *NDN* are maternally imprinted (paternally expressed) genes [42].

Comparing expression levels across all four evaluated groups (early onset cases, preterm controls, late onset cases, and term controls), the lowest expression was observed in early onset cases, and the highest expression was observed in preterm controls for both *MEST* and *NDN*. This elevated expression in the preterm controls increases the contrast in expression, likely contributing to the observed significant difference in expression for these two genes among early onset cases. While we detected no significant gene expression differences between the preterm and term controls (Appendix A), this downregulation in expression from earlier to later gestational stages could suggest a natural dynamic change in expression across gestation that is dysregulated with early onset of preeclampsia. In this case, elevated expression of *MEST* and *NDN* early in gestation tapers off toward later stages of gestation among uncomplicated pregnancies. However, this elevated baseline is not realized in early onset preeclampsia. Meanwhile, in later onset preeclampsia, this dysregulation in expression early in gestation is not triggered, and any temporal changes in *MEST* and *NDN* expression across gestation are not affected by later onset preeclampsia.

The sex-stratified analyses suggest that the downregulation in *MEST* expression comparing early onset cases and preterm controls was restricted to female placentas (Figure 3B); however, a similar trend, albeit not significant, was also apparent in the comparison between early onset cases and preterm controls in male placentas (Figure 3C). This suggests that while the association is more robust due to lower intragroup variance among female placentas in our study population, the observed downregulation in *MEST* expression is generally consistent across male and female placentas.

While few studies have linked *NDN* to pregnancy complications, the literature does support the involvement of *MEST* aberrations in various pregnancy-related outcomes, including in vitro fertilization (IVF)/intracytoplasmic sperm injections (ICSIs) [43], viral load [44], gestational diabetes [45], and preeclampsia [46]. Interestingly, fetal growth restriction is among the most commonly reported outcomes linked to deviations in *MEST* activity [44,47,48,49,50], likely at least partially attributable to the purported role of *MEST* in placental angiogenesis and implantation [34]. In studies where placental *MEST* gene expression levels were assessed [44,47,48], expression was generally downregulated, consistent with our findings in the current study. One study evaluated aberrations in *MEST* cord blood methylation levels in relation to preeclampsia and reported upregulated methylation in cases compared to controls [46].

Given the commonalities in placental assessments across multiple pregnancy complications, changes in *MEST* expression and/or methylation are likely not a suitable marker to characterize preeclampsia-specific disease trajectory. However, the consistency observed across the reported findings does suggest the possible role of this gene as an intrauterine biosensor that captures pathological deviations from appropriate placental development.

At least one other study performed a genome-wide survey of imprinted gene expression in relation to preeclampsia status [27]. Using microarray data, this study assessed correlations between each gene and two pregnancy outcomes, fetal growth restriction and preeclampsia. Subsequently, known imprinted genes within the microarray were tested to assess whether correlations with the two outcomes of interest were enriched within this subset of genes. Indeed, the findings suggest greater dysregulation among imprinted genes specifically in relation to preeclampsia than would be expected by chance. The study also reports an overall trend of downregulation among paternally expressed imprinted genes, a finding that is consistent with the downregulation of the paternally expressed *MEST* and *NDN* genes observed in the current study.

Restricting microarray data to known and putative imprinted genes, Zadora et al. reported upregulation of the imprinted gene *DLX5* among preeclampsia cases compared to controls, with a more pronounced effect among early onset cases [26]. While not reaching statistical significance based on the stringent threshold applied, our study also indicated *DLX5* upregulation (FDR = 0.15) comparing early onset cases to preterm controls (Appendix A).

Several limitations of the study warrant consideration. While the sample size was on par with previously reported studies assessing gene expression alterations between preeclampsia cases and controls, the study was underpowered to conduct sensitivity analyses to more comprehensively evaluate the influence of potential confounding factors in our study. For example, while we conducted sensitivity analyses to evaluate the impact due to the onset of labor, additional differences in delivery mode (e.g., caesarean vs. vaginal) may also warrant further evaluation to add rigor to our findings. However, since we did not observe differences in the distribution of delivery mode across preeclampsia cases and controls, we do not expect a differential impact across the groups. Similarly, while we performed sex-stratified analyses, we cannot rule out that our finding of a significant difference within the female subset and a nonsignificant difference within the male subset may be due to limitations in sample size. Additionally, the normotensive control subjects selected as comparison groups for the respective early onset and late onset preeclampsia cases may have implications for our results, specifically with respect to the preterm controls used in the comparison against the early onset cases. Unlike the term controls, preterm controls do not stem from uncomplicated pregnancies, since preterm labor itself is a pregnancy complication. Despite contrasting the early onset cases with preterm normotensive controls to better disentangle preeclampsia-specific effects from preterm-related effects, we cannot rule out the possibility that unique characteristics of the selected preterm controls may contribute to the observed significant differences in expression among early onset cases. Finally, our findings point to altered transcriptional activity of genes in placental biopsies. Based on these findings alone, we are not able to substantiate the functional consequences of these alterations with respect to implications for translational outputs. Additionally, the imprinted genes were assessed to capture gene-wide expression patterns. It is possible that observed differences in *MEST* expression, for example, are due to differences in specific relevant isoforms, which are not able to be discerned in the current study. Furthermore, given that the placenta is a composite of multiple cell types, we cannot rule out that the observed differences in transcript levels may, in fact, reflect a shift in cell-type composition across cases and controls. Additional studies that further interrogate and characterize the implications for the observed gene expression differences are, therefore, warranted.

This study represents one of the most comprehensive evaluations of imprinted gene expression contrasting preeclampsia and non-preeclampsia placentas delivered preterm and at term. Our findings suggest downregulation of placental *MEST* and *NDN* expressions among preterm preeclampsia placentas. While not a preeclampsia-specific biomarker, downregulation of placental *MEST* expression may be a biosensor of adverse pregnancy outcomes. Additional studies expanding on this paradigm, including evaluations of additional pregnancy-related outcomes and sensitivity to environmental agents that may trigger these altered trajectories, are warranted.

## Figures and Tables

**Figure 1 genes-11-01146-f001:**
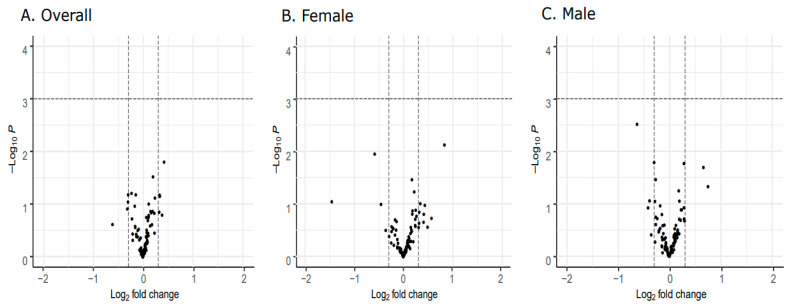
Differential placental imprinted gene expression analyses between late onset cases and term controls. Volcano plots depict log2 fold change values between cases and controls on the x-axis and -log10 p-values on the y-axis. Points falling above the dashed horizontal line indicate genes that are significantly differentially expressed on the basis of a false discovery rate (FDR) < 0.05. (**A**) No genes are differentially expressed between cases and controls. In the sex-stratified analyses, no genes are differentially expressed comparing female cases and controls (**B**) or male cases and controls (**C**).

**Figure 2 genes-11-01146-f002:**
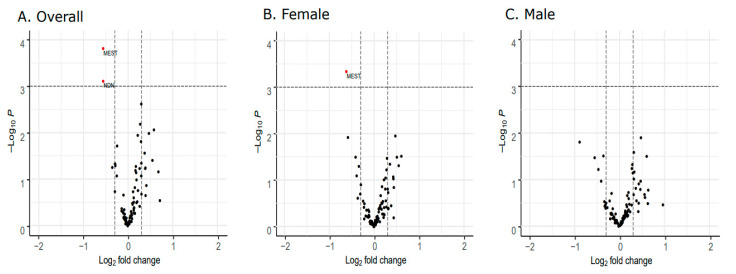
Differential placental gene expression analyses between early onset cases and preterm controls. Volcano plots depict log2 fold change values between cases and controls on the x-axis and -log10 p-values on the y-axis. Points (in red) falling above the dashed horizontal line indicate genes that are significantly differentially expressed on the basis of an FDR <0.05. (**A**) Two genes, *MEST* and *NDN*, are significantly downregulated among cases compared to controls. In the sex-stratified analyses, *MEST* is also significantly downregulated comparing cases and controls within the female group (**B**). No significant difference in gene expression between cases and controls is observed within the male group (**C**).

**Figure 3 genes-11-01146-f003:**
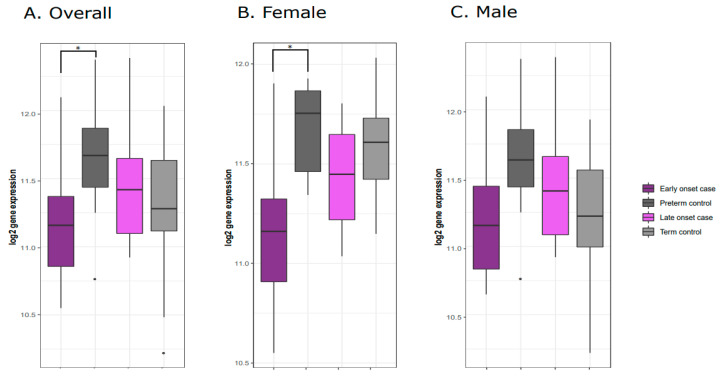
*MEST* gene expression levels across preeclampsia cases and controls. (**A**) Overall downregulation of *MEST* gene expression levels is observed comparing early onset preeclampsia cases and preterm controls. Similar trends in *MEST* downregulation are observed in analyses restricted to female (**B**) and male (**C**) placenta.

**Figure 4 genes-11-01146-f004:**
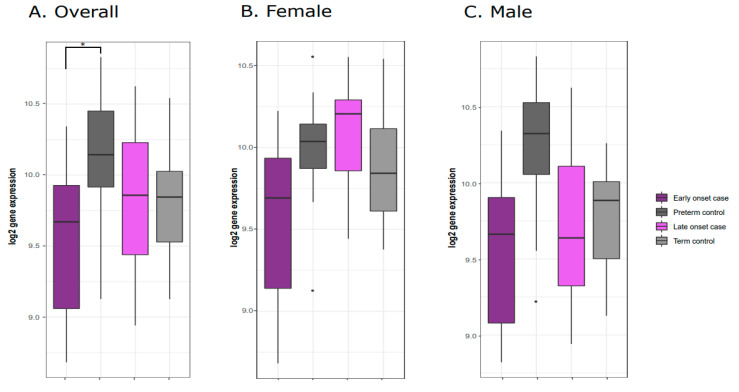
*NDN* gene expression levels across preeclampsia cases and controls. (**A**) Overall downregulation of NDN gene expression levels is observed comparing early onset cases to preterm controls. Similar trends in *NDN* downregulation are observed in analyses restricted to female (**B**) and male (**C**) placenta.

**Table 1 genes-11-01146-t001:** Demographic characteristics across preeclampsia and non-preeclampsia subjects included in the study (*n* = 99).

Variable	Early Onset Cases (*n* = 24)	Preterm Controls (*n* = 19)	*p*-Value	Late Onset Cases (*n* = 25)	Term Controls (*n* = 31)	*p*-Value *
	Mean (SD)	Mean (SD)		Mean (SD)	Mean (SD)	
Gestational age (weeks)	32.4 (1.5)	34.6 (3.1)	<0.01	39.0 (1.4)	39.2 (0.9)	0.39
Birthweight (grams)	1645.5 (508.6)	2351.8 (651.1)	<0.01	3099.1 (547.9)	3312.2 (405.0)	0.10
Maternal age (years)	27.0 (6.8)	26.9 (5.6)	0.96	26.9 (5.9)	25.3 (4.8)	0.27
	N (%)	N (%)		N (%)	N (%)	
Delivery method:						
Caesarean	14 (58.3)	7 (36.8)	0.27	8 (32.0)	6 (19.4)	0.44
Vaginal	10 (41.7)	12 (63.2)		17 (68.0)	25 (80.6)	
Infant sex:						
Female	11 (45.8)	9 (47.4)	1.00	9 (36.0)	12 (38.7)	1.00
Male	13 (54.2)	10 (52.6)		16 (64.0)	19 (61.3)	
Race/ethnicity:						
White	18 (75.0)	6 (31.6)	0.01	19 (76.0)	25 (80.6)	0.93
Non-white	6 (25.0)	13 (68.4)		6 (24.0)	6 (19.4)	
Parity:						
Nulliparous	18 (75.0)	12 (63.2)	0.61	22 (88.0)	26 (83.9)	0.96
Parous	6 (25.0)	7 (36.8)		3 (12.0)	5 (16.1)	

* *p*-Values by Student’s *t*-test for continuous variables and Chi-squared test for categorical variables.

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
