# Peer review of "Differences in Placental Imprinted Gene Expression across Preeclamptic and Non-Preeclamptic Pregnancies"

_genes, 2020, doi:10.3390/genes11101146_

Round 1

Reviewer 1 Report

This manuscript investigated the expression of imprinted gene between preeclamptic and non-preeclamptic placentas. Preeclamptic placentas were separated between early onset and late onset and each subset was aged-matched with normotensive placentas. There were no differences in demographics features between late onset and term controls. Expected differences in gestational age and birth weight were observed between early onset preeclampsia and preterm birth.

No differentially expressed genes were found between the two control groups and between late onset preeclamptic placentas and term controls. Two genes, NDN and MEST, were downregulated in early onset placentas compared to pre-term controls. Only MEST showed significant downregulation in female but not male placentas in a sex-stratified analysis, although a down-regulation trend was observed in male placenta.

POSITIVE OBSERVATIONS

The study identified separate aged-matched controls for the early and late onset cases, although some data might suggest a skewed expression in the preterm control cases (see major point below).

This is a focused study with a thought-through design, in particular related to the control cases for early onset preeclampsia. The criteria for enrolment, and diagnosis of preeclampsia were well defined. The number of cases in each subset was relatively small but on par with similar studies.   The results on MEST agree with previous data on other placenta complication and therefore not preeclampsia-specific. Data on NDN are sparser and represent a more novel observation that will need further validation.

The authors well present some of the limitations of the study and do not over-sell the results.

Major critics:

Interestingly, the lowest expression of MEST and NDN was observed in early onset cases and the highest expression in preterm controls.  The reviewer acknowledge that the control cases for early onset preeclampsia are notoriously difficult to validate as preterm birth, even if normotensive, have other often unknown issues that caused preterm labor, and appreciate the use of aged-matched pre-term cases as valid control group. However, as the authors themselves point out in the discussion, these higher expression in preterm controls contributes to the significance in gene expression. The authors attempt to give an explanation to the lower expression in early onset cases but fail to explain the upregulation in preterm control cases. It is not clear how the investigators would validate that the findings on MEST and NDN are not simply a consequence of the chosen control and the skewed data.  

Minor revision:

155-157: the manuscript states that “while the overall study population is predominantly (>75%) Caucasian, the majority of early onset cases are African American (Table1).” If the reviewer has correctly read the table, the majority of early onset seems to be white (18 vs 6 non-white) whereas the majority of the preterm control cases were non-white (13 vs 6 white).

185: NDN is incorrectly reported as NECTIN instead of NECDIN

Supplemental figure S1 is not mentioned in the manuscript and the rest of the supplemental figures are reported incorrectly (173: Suppl. Fig 1 is instead Suppl. Fig 2; 175: Suppl. Fig 2 is instead suppl Fig. 3 etc).

In the Vulcano plots, a different background color in the significant portion of the plot would help the reader to identify the area to pay attention to.

Author Response

Comments from Review #1:

Point 1:

Interestingly, the lowest expression of MEST and NDN was observed in early onset cases and the highest expression in preterm controls.  The reviewer acknowledge that the control cases for early onset preeclampsia are notoriously difficult to validate as preterm birth, even if normotensive, have other often unknown issues that caused preterm labor, and appreciate the use of aged-matched pre-term cases as valid control group. However, as the authors address themselves point out in the discussion, these higher expression in preterm controls contributes to the significance in gene expression. The authors attempt to give an explanation to the lower expression in early onset cases but fail to explain the upregulation in preterm control cases. It is not clear how the investigators would validate that the findings on MEST and NDN are not simply a consequence of the chosen control and the skewed data.  

Indeed, the reviewer raises a legitimate concern.  We sought to discern potential underlying differences within the selected preterm controls by comparing them to the uncomplicated, term controls and noted no differences in imprinted gene expression patterns (Figure S1).  One possibility for the upregulated expression observed in the preterm controls is that it reflects a natural dynamic trend in expression across gestation.  As in, MEST and NDN expression are naturally elevated in early gestation (as captured by the preterm controls), and this level of expression attenuates closer to term.  We attempt to clarify this possible rationale in the discussion section (Line 228-229).

Still, it is possible that there is something unique to this set of controls that fundamentally drives the observed observations of significant upregulated MEST and NDN expression among early onset cases.  We highlight this limitation in the discussion section (Lines 287-291).

Point 2:

155-157: the manuscript states that “while the overall study population is predominantly (>75%) Caucasian, the majority of early onset cases are African American (Table1).” If the reviewer has correctly read the table, the majority of early onset seems to be white (18 vs 6 non-white) whereas the majority of the preterm control cases were non-white (13 vs 6 white).

We thank reviewer for catching this typographical error.  We have revised the manuscript to correctly attribute the race/ethnicity distributions in our study population as follows (Line 167-169): “While the overall study population is predominantly (>75%) White, the majority of preterm controls (68.4%) are Non-white”

Point 3:

185: NDN is incorrectly reported as NECTIN instead of NECDIN

We thank reviewer for catching this typographical error.  We now correctly refer to Necdin in the manuscript (Line 186)

Point 4:

Supplemental figure S1 is not mentioned in the manuscript and the rest of the supplemental figures are reported incorrectly (173: Suppl. Fig 1 is instead Suppl. Fig 2; 175: Suppl. Fig 2 is instead suppl Fig. 3 etc).

We thank reviewer for catching this error. The reference to Supplemental Figure 1 has been corrected to Figure S1 (Lines 160 and 227) and all other Supplemental Figure labels have also been updated to reflect the correct labeling [Figure S2 (Line 177), Figure S3 (Line 179), Figure S4 (Line 205), Figure S5 (Line 206), Figure S6 (Line 270)].

In the Vulcano plots, a different background color in the significant portion of the plot would help the reader to identify the area to pay attention to.

To highlight relevant regions of the volcano plots, we have added the following statement to the figure captions for Figures 1, 2, S2, S3, S4 and S5: “Points falling above the dashed horizontal line indicate genes that are significantly differentially expressed based on an FDR < 0.05”.

Reviewer 2 Report

The manuscript by Deyssenroth et al. presents a study of expression of imprinted placental genes early and late onset preeclampsia cases using preterm and term normotensive placentas as controls. Whereas no differences in gene expression were observed between late onset preeclampsia, the authors report a downregulation of placental Mesoderm-specific transcript (MEST) and Necdin (NDN) genes in early onset preeclampsia compared controls.

In agreement with previous reports, a potential role for placental MEST expression as a marker of an adverse in utero environment is proposed.

The authors define early and late onset preeclampsia with a cutoff of deliveries at 37 weeks of gestation, however current definitions from the ACOG and ISSHP, define it as delivery at 34 weeks (Obstetrics & Gynecology 2013 122(5):1122; Pregnancy Hypertension: An International Journal of Women’s Cardiovascular Health 2014 4(2):97). This should be mentioned in Methods and added to discussion.

Limitations related to the sample size, that may restrict sensitivity analyses, as well as the chosen control groups, in particular the use of preterm control versus early onset preeclamptic group, are acknowledged and discussed by the authors. The potential effects on the results obtained of performing the studies with a population of women that is overweight and obese should be discussed.

Values of p considered to be statistically significant should be reported in Statistical Analyses section.

Figures 1 and 2 should indicate that these are volcano plots. Legend of Figure 2 should indicate that the genes down regulated (MEST and NDN) compared to controls are indicated as red dots in graph, also panels B and C should be labelled Males and Females, respectively.

Similarly, in Figures 3 and 4 panels B and C should be labelled Males and Females respectively.

Format of references needs to be standardized, some citations present the full journal's name other abbreviations, similarly with vol, issue number etc. 

Author Response

Point 1:

The authors define early and late onset preeclampsia with a cutoff of deliveries at 37 weeks of gestation, however current definitions from the ACOG and ISSHP, define it as delivery at 34 weeks (Obstetrics & Gynecology 2013 122(5):1122; Pregnancy Hypertension: An International Journal of Women’s Cardiovascular Health 2014 4(2):97). This should be mentioned in Methods and added to discussion.

Indeed, the ACOG and ISSHP currently define early onset preeclampsia as cases that were diagnosed before 34 weeks gestation, with delivery typically occurring in the days/week following diagnosis. This discrimination is based largely on clinical outcome with severe form of preeclampsia more common and virtually all infants born after 34 weeks ‘ surviving.

The research definition we used to delineate early onset from late onset cases in the current study was motivated by pathophysiological differences that become more apparent by 37 weeks’ gestation.  By the  37 week cut-point, preeclampsia is not associated with an increased risk of fetal growth restriction, and there are fewer vascular changes in the placenta. Nonetheless, in the presented analysis, the gestational age at delivery among early onset cases ranged from 30-35 weeks gestation, with all but 2 deliveries occurred within 34 weeks gestation (and disease onset likely occurring prior to hospital admission).  The gestational age at delivery among late onset cases ranged from 37-41 weeks gestation.  Hence, the early and late onset cases in the current study fall within the cut-point definitions outlined by the ACOG.  We have added this clarification in the Methods section of the manuscript (Lines 100-109).

Point 2:

Values of p considered to be statistically significant should be reported in Statistical Analyses section.

We have expanded our reference to the implemented false discovery rate cut-offs with the following (Line 155-156):  “Significant differences in gene expression were determined based on p-values adjusted for multiple comparisons using a false discovery rate (FDR) value < 0.05.

Point 3:

Figures 1 and 2 should indicate that these are volcano plots. Legend of Figure 2 should indicate that the genes down regulated (MEST and NDN) compared to controls are indicated as red dots in graph, also panels B and C should be labelled Males and Females, respectively.

Similarly, in Figures 3 and 4 panels B and C should be labelled Males and Females respectively.

The individual panels in Figures 1-4 are now labelled to indicate whether they reflect the overall, female-restricted or male-restricted analyses.  All figures (Figures 1,2, S2-S5) that depict volcano plots now include the following in the captions: “Volcano plots depicting log2 fold change values on the x-axis and -log10 p-values on the y-axis.  Points (in red) falling above the dashed horizontal line indicate genes that are significantly differentially expressed based on an FDR < 0.05.”

Point 4:

Format of references needs to be standardized, some citations present the full journal's name other abbreviations, similarly with vol, issue number etc. 

The Reference section has been revised to be more consistent with respect to journal abbreviations and volume/issue number.

Reviewer 3 Report

Deyssenroth et al. investigated placental imprinted gene expression across early- and late-onset pre-eclamptic placentas and their respective controls.  This is a well-written, well-designed study.  The sample numbers are high enough (i.e. 24 for EO-PE, 19 for PTC, 25 for LO-PE and 31 for TC), and it is the first study that selectively examines the expression of imprinted genes in both early- and late-onset PE.  The study reports a down-regulation of two imprinted genes, MEST and NDN, in the early-onset PE samples only.  There is also a trend towards increased expression of DLX5 in EO-PE.  This study validates previous reports of reduced NDN and increased DLX5 in PE, and it newly identifies MEST as another player.  There are several issues, which should be addressed:

  • Where are MEST and NDN localised in the placenta? To substantiate the results of this study, it would be good to validate the gene changes at the protein level, ideally by IHC, which would indicate the cellular localisation of MEST and NDN in the 4 groups of samples. Additional protein quantification could also be performed by western blotting.
  • According to Figs. 2b and 2c, sex-specific differences are also evident for NDN, not only for MEST, i.e. NDN is no longer significantly downregulated in the male or female cohort alone. The sex specific analysis is displayed twice for each gene, i.e. Fig. 2b-c and 3b-c for MEST, and Fig. 2b-c and 4b-c for NDN.  However, only Fig. 4b-c is used to discuss the effect of sex on NDN, Fig. 2b-c being ignored.  Can you please explain.
  • The authors claim that while the overall study population was predominantly (>75%) Caucasian, the majority of early onset cases were African American. In fact, according to Table 1, it is the majority of the pre-term controls that were African American (68.4%), whilst the EO-PE were only 25% African American.  Please check/correct.

Minor:

The plural of placenta is placentas or placentae, not placenta

Author Response

Point 1:

Where are MEST and NDN localised in the placenta? To substantiate the results of this study, it would be good to validate the gene changes at the protein level, ideally by IHC, which would indicate the cellular localisation of MEST and NDN in the 4 groups of samples. Additional protein quantification could also be performed by western blotting.

Indeed, additional readouts of MEST and NDN, including the translational end products of the assessed mRNA transcriptional activity, would further substantiate the relevance of these genes as markers of early onset preeclampsia.  In addition, cell-type specific localization of these markers could also further our understanding of the mechanistic underpinnings driving the pathophysiology of early onset preeclampsia.  While valuable, we were limited in the availability of the relevant biospecimens to conduct these additional assessments, which necessarily restricted the scope of the current study to the identification of potential placental biomarkers distinguishing early onset and late onset preeclampsia cases from controls.  We have added to the discussion section the need for additional studies that further interrogate and characterize the potential role of these markers in the pathophysiology of disease. (Lines 291-298)

Point 2:

According to Figs. 2b and 2c, sex-specific differences are also evident for NDN, not only for MEST, i.e. NDN is no longer significantly downregulated in the male or female cohort alone. The sex specific analysis is displayed twice for each gene, i.e. Fig. 2b-c and 3b-c for MEST, and Fig. 2b-c and 4b-c for NDN.  However, only Fig. 4b-c is used to discuss the effect of sex on NDN, Fig. 2b-c being ignored.  Can you please explain.

Indeed, the overall effect of NDN downregulation among early onset cases compared to controls does not reach statistical significance in either of the sex-stratified subgroups (Figures 2b and 2c).  This is likely driven by sample size considerations – given the reduced sample size in each subgroup, we are likely underpowered to detect significant differences.  We now additionally reference Figures 2b and 2c to underscore this point in the Results with respect to the NDN sex-stratified analysis (Line 196).

Point 3:

The authors claim that while the overall study population was predominantly (>75%) Caucasian, the majority of early onset cases were African American. In fact, according to Table 1, it is the majority of the pre-term controls that were African American (68.4%), whilst the EO-PE were only 25% African American.  Please check/correct.

We thank reviewer for catching this typographical error.  We have revised the manuscript to correctly attribute the race/ethnicity distributions in our study population as follows (Line 167-169): “While the overall study population is predominantly (>75%) White, the majority of preterm controls (68.4%) are Non-white”

Point 4:

The plural of placenta is placentas or placentae, not placenta

We thank reviewer for catching this typographical error.  We have revised the manuscript to reflect the plural form of the word as appropriate (Lines, 25, 126, 179, 236, 238, 239, 241, 300, 302, 311)

Round 2

Reviewer 3 Report

The authors responded to all points raised by myself and the other reviewer.  I recommend the manuscript for publishing at Genes.

Author Response

Thank you.